# Docking sites inside Cas9 for adenine base editing diversification and RNA off-target elimination

Shuo Li[1], Bo Yuan [2], Jixin Cao[3], Jingqi Chen [3], Jinlong Chen[4], Jiayi Qiu[4], Xing-Ming Zhao[3], Xiaolin Wang[1,5✉], Zilong Qiu [2,6✉] & Tian-Lin Cheng [4✉]

Base editing tools with diversified editing scopes and minimized RNA off-target activities are required for broad applications. Nevertheless, current *Streptococcus pyogenes* Cas9 (SpCas9)-based adenine base editors (ABEs) with minimized RNA off-target activities display constrained editing scopes with efficient editing activities at positions 4-8. Here, functional ABE variants with diversified editing scopes and reduced RNA off-target activities are identified using domain insertion profiling inside SpCas9 and with different combinations of TadA variants. Engineered ABE variants in this study display narrowed, expanded or shifted editing scopes with efficient editing activities across protospacer positions 2-16. And when combined with deaminase engineering, the RNA off-target activities of engineered ABE variants are further minimized. Thus, domain insertion profiling provides a framework to improve and expand ABE toolkits, and its combination with other strategies for ABE engineering deserves comprehensive explorations in the future.

[1] Department of interventional Radiology, Zhongshan Hospital, Fudan University, 180 Fenglin Road, 200032 Shanghai, China. [2] Institute of Neuroscience, State Key Laboratory of Neuroscience, CAS Center for excellence in Brain Science and Intelligence Technology, Chinese Academy of Sciences, 200031 Shanghai, China. [3] Institute of Science and Technology for Brain-Inspired Intelligence, Key Laboratory of Computational Neuroscience and Brain-Inspired Intelligence, Ministry of Education, Fudan University, 200433 Shanghai, China. [4] Institute for Translational Brain Research, MOE Frontiers Center for Brain Science, Fudan University, 200032 Shanghai, China. [5] Shanghai Institute of Medical Imaging, 200032 Shanghai, China. [6] National Clinical Research Center for Aging and Medicine, Huashan Hospital, Fudan University, Shanghai, China. ✉email: wang.xiaolin@zs-hospital.sh.cn; zqiu@ion.ac.cn; chengtianlin@fudan.edu.cn

Base editing tools, composed of ABEs mediating adenine to guanine (A–G) conversion and CBEs mediating cytosine to thymine (C–T) conversion without inducing double-strand breaks (DSBs), are powerful tools for targeted nucleotide editing in genomic DNA[1–3]. Theoretically, ~47% disease-associated single-nucleotide polymorphisms (SNPs) are G·C to A·T mutations, which could be corrected by ABEs[4], making ABEs promising tools for gene therapy.

Diversified editing scopes will enable ABEs to access more target nucleotides and thus expand their capabilities for the correction of disease-associated SNPs. As single-stranded DNA (ssDNA) substrates are required for effective editing, the accessibility of ssDNA loop to deaminases is critical for the diversification of editing scopes, in addition to the features of deaminases and protospacer adjacent motif (PAM) of CRISPR-Cas systems[3,5–10]. Generally, existing ABEs are mainly generated through "head-to-tail" fusion strategy, which means adenosine deaminases are fused to either the amino terminus (N-terminus) or the carboxy terminus (C-terminus) of Cas nickase (nCas) or catalytically inactive Cas (dCas) proteins[3]. As the N-/C-termini of Cas proteins are relatively fixed, it often leads to constrained deaminase locations and restricted editing scopes (generally ~4–5 nucleotides within protospacer). Recently, circular permutation has been deployed for SpCas9 termini repositioning and several active SpCas9 circular permutants (Cas9-CPs) with altered N-/C-termini have been established[11]. As altered N-/C-termini bring significant changes to deaminase locations, the derived Cas9-CP-ABEs display expanded editing scopes[6,12]. Additionally, variant adenosine deaminase mutants have been generated and derived ABE variants display superior activities and expanded editing scopes[12,13]. Nevertheless, the editing scopes of derived ABE variants remain constrained, and alternative strategies are needed to further diversify the editing scopes.

Moreover, safety concerns have been raised about the off-target activities for ABEs, which may limit future applications[14–17]. One major issue is that original ABEs could induce obvious off-target mutations at transcriptome level[14–17]. Though ABE variants with minimized RNA off-target activities have been developed through deaminase engineering, their editing scopes were severely restricted, with editing windows mainly at positions 4–8 (the PAM was counted as 21–23 unless otherwise indicated). Therefore, there is an urgent need to develop ABE variants with diversified editing scopes and minimized RNA off-target activities.

It has been demonstrated that the structure of SpCas9 protein is flexible enough to permit functional domain insertion without interfering its DNA binding ability[18]. The relative distances of docking sites (DSs) inside Cas9 to ssDNA loop are more variable as compared to N-/C-termini, and domain insertion inside Cas9 may limit the excessive flexibility of functional domains, so it is possible that domain insertion could achieve scope diversifications and off-target minimization simultaneously.

Here we initially screen for functional ABE variants using domain insertion profiling across 24 potential DSs in SpCas9 nickase (nSpCas9, representing D10A nickase unless otherwise specified), which are distributed across different domains of Cas9, with the majority in discrete regions and in close to ssDNA loop[19]. The identified functional ABE-nSpCas9-DS variants retain high editing activities and display diversified editing scopes, with adenine nucleotides at protospacer positions 2–16 now targetable. Indeed, several engineered ABE-nSpCas9-DS variants such as ABE-nSpCas9-DS770 display obvious shifted editing scopes with maximal activities at positions 9–16, which has not been achieved previously. Transcriptome analysis further demonstrate that ABE-nSpCas9-DS variants display significantly reduced RNA off-target activities as compared to N-terminal counterparts, and RNA off-target activities could even be eliminated by domain insertion in combined with engineered deaminases. Additionally, in consistent with previous studies[20], we also noticed that several functional ABE-nSpCas9-DS variants display obvious cytosine deamination activities and the editing scopes display a DS-dependent manner. Taken together, our study provides a series of ABE toolkits with diversified editing scopes and improved security. We also confirm that domain insertion is a powerful strategy and its combination with other strategies for ABE engineering deserves comprehensive investigations in the future.

## Results

**Diversified targeting scopes of ABE variants with domain insertion profiling.** Existing ABE tools were initially developed by fusing TadA-TadA* heterodimer to the N-terminus of nCas9 (D10A), in which TadA served as a scaffold for substrate without ssDNA deamination activity while TadA* is an evolved adenosine deaminase with obvious ssDNA deamination activity. Considering that insertion inside nSpCas9 would lead to domain repositioning and TadA may interfere with deamination process, we initially chose TadA*-TadA* homodimer to screen for functional ABE variants using domain insertion profiling. In all, 24 potential DSs (Table 1, selection criteria in Supplementary Discussion), distributed across different domains of Cas9, mainly in discrete and flexible regions and close to ssDNA loop[19] (Fig. 1a, b), were selected. Their activities were evaluated against one single guide RNA (sgRNA) containing multiple adenines across the 20-nt protospacer (sgRNA-1) (Supplementary Fig. 1).

Sanger sequencing and quantification with EditR software[21] revealed that 11/24 ABE (TadA*-TadA*)-nSpCas9-DS variants (535-/583-/770-/793-/801-/895-/905-/915-/1010-/1029-/1249-TadA*-TadA*) displayed ≥20% A–G conversion efficiency at

**Table 1 Location summary of functional DSs for ABE-nSpCas-DS variants.**

| Name | Domain at DS site | Start site after functional domain | Sequence around DS site |
|---|---|---|---|
| nCas9-DS$^{535}$ | REC lobe | 535 | VTEGM/RKPAF |
| nCas9-DS$^{583}$ | REC lobe | 583 | VEISG/VEDRF |
| nCas9-DS$^{770}$ | RuvC-II | 770 | RENQT/TQKGQ |
| nCas9-DS$^{793}$ | HNH | 793 | IKELG/SQILK |
| nCas9-DS$^{801}$ | HNH | 801 | LKEHP/VENTQ |
| nCas9-DS$^{895}$ | HNH | 895 | KLITQ/RKFDN |
| nCas9-DS$^{905}$ | HNH | 905 | LTKAE/RGGLS |
| nCas9-DS$^{919}$ | HNH | 919 | AGFIK/RQLVE |
| nCas9-DS$^{1010}$ | RuvC-III | 1010 | ESEFV/YGDYK |
| nCas9-DS$^{1029}$ | RuvC-III | 1029 | KSEQE/IGKAT |
| nCas9-DS$^{1249}$ | CTD | 1249 | KLKGS/PEDNE |

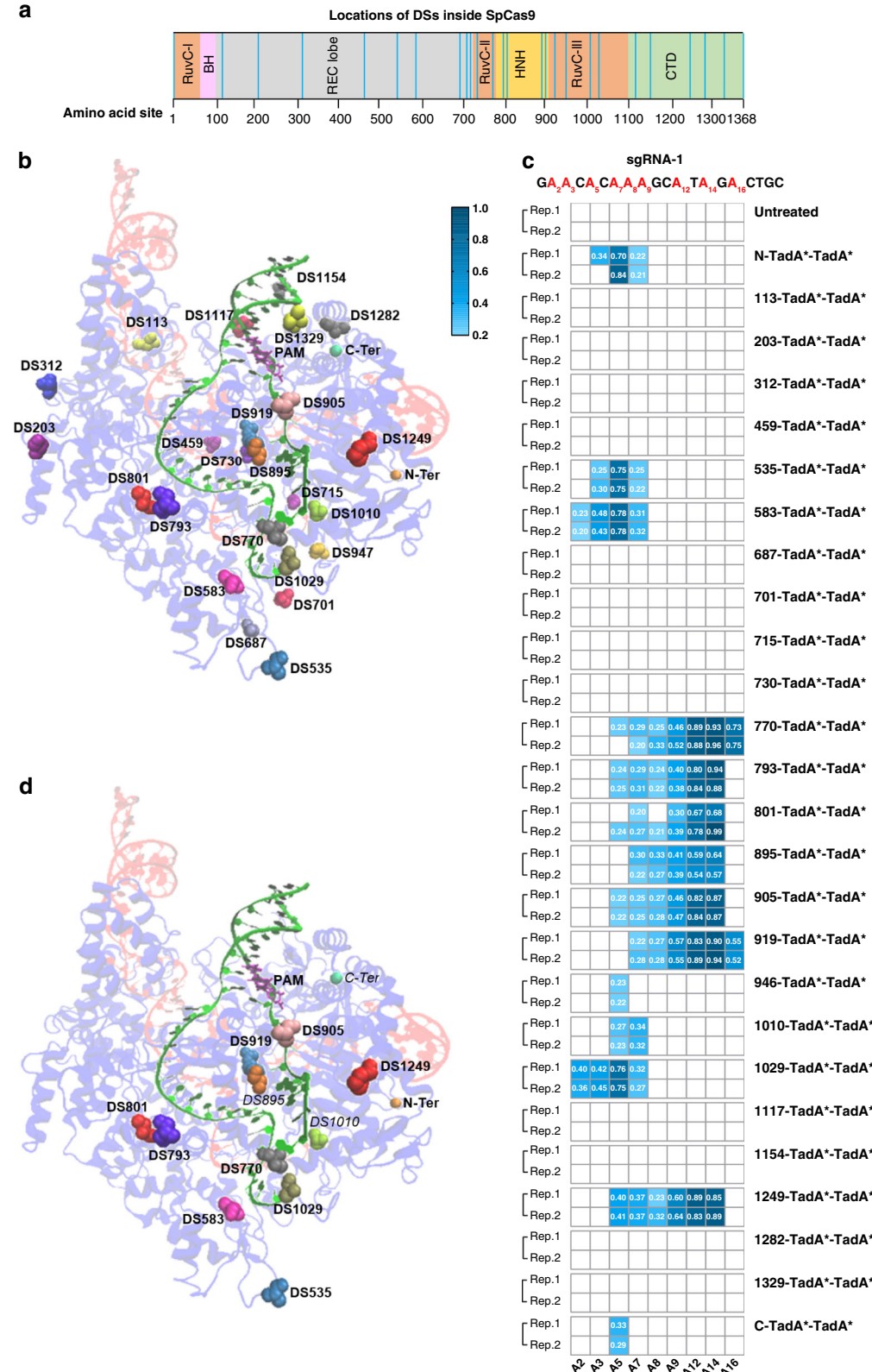

**Fig. 1 Screening for functional ABE (TadA*-TadA*)-nSpCas9-DS variants. a** Diagram showing SpCas9 domain organization and 24 DSs (light blue lines) screened in this study. **b** Locations of 24 DSs and N-/C-termini in the structure of SpCas9:sgRNA:DNA complex (PDB:5F9R). **c** Base editing activities of ABE (TadA*-TadA*)-nSpCas9-DS variants against sgRNA-1. Adenines in sgRNA-1 were labeled as red. A–G conversion frequencies at every adenine nucleotide in 20 bp protospacer were quantified with EditR and shown in heat map. Two independent experiments were performed and editing frequencies >0.20 were labeled in the heat map. **d** Locations of active DSs for ABE variants and N-/C-termini in the structure of SpCas9:sgRNA:DNA complex (PDB:5F9R). DS895, DS1010 and C-Ter were shown in italic type because of lower activities of derived ABE variants as compared to other DSs. Source data are available in the Source data file.

specific adenine nucleotide(s) (Fig. 1c, d), with 10/11 displaying ≥40% A–G conversion efficiency and 9/11 (535-/583-/770-/793-/801-/905-/915-/1029-/1249-TadA*-TadA*) displaying comparable activities to N-terminal counterpart. It was revealed that functional ABE (TadA*-TadA*)-nSpCas9-DS variants could be classified into two major groups according to their maximal editing sites, with 535-/583-/1029-TadA*-TadA* displaying maximal activities at A5, similar to original ABE tools, while 770-/793-/801-/895-/905-/915-/1249-TadA*-TadA* displaying maximal activities at A12–14, representing a group of ABE variants with robustly shifted editing scopes not reported previously (Fig. 1c).

We wondered whether TadA*-TadA* dimer insertion would interfere with SpCas9's DNA cleavage activity. ABE (TadA*-TadA*)-SpCas9-DS variants were generated and their DNA cleavage activities were assessed against two sgRNAs. It was revealed that insertions at different DSs have distinct impact on SpCas9's DNA cleavage activity (Supplementary Fig. 2) and no obvious correlation was observed between the base editing activities of ABE (TadA*-TadA*)-nSpCas9-DS variants and the DNA cleavage abilities of corresponding ABE (TadA*-TadA*)-SpCas9-DS counterparts. For example, ABE (TadA*-TadA*)-nSpCas9-DS770-919, with insertions around and inside HNH domain, displayed robust A–G conversion activities, while their corresponding ABE (TadA*-TadA*)-SpCas9-DS770-919 variants lost DNA cleavage activities because of HNH nuclease disruption (Supplementary Fig. 2). Additionally, several ABE (TadA*-TadA*)-SpCas9-DS variants showed site-dependent DNA cleavage activities. For example, ABE (TadA*-TadA*)-SpCas9-DS1029 and -DS1117 displayed comparable cleavage activities against sgRNA-1 while their activities against sgRNA-E8 decreased significantly as compared to wild-type SpCas9 (Supplementary Fig. 2).

We next investigated which TadA* monomer deaminated adenine nucleotide(s) in functional ABE-nSpCas9-DS variants by introducing loss-of-function (LOF) mutations (H57A/E59A) into either TadA* monomer. We generated 10 ABE (TadA* (LOF)-TadA*)-nSpCas9-DS and 10 ABE (TadA*-TadA* (LOF))-nSpCas9-DS variants for A–G conversion analysis against sgRNA-1. Overall, the former TadA* deaminase activity is almost dispensable as all ABE (TadA* (LOF)-TadA*)-nSpCas9-DS variants displayed comparable editing activities and slightly narrower editing scopes as compared to ABE (TadA*-TadA*)-nSpCas9-DS counterparts (Fig. 2a). In contrast, the latter TadA* deaminase activity is critical as the activities of ABE (TadA*-TadA* (LOF))-nSpCas9-DS variants were severely impaired, with 583-/793-/895-/905-/919-TadA*-TadA* (LOF) losing almost all activities and 535-/770-/801-/1029-/1249-TadA*-TadA* (LOF) displaying modest/lower activities and obviously narrower editing scopes (Fig. 2b). As both TadA* monomers inside nSpCas9 could serve as scaffold for ssDNA loop, above results indicated that the former TadA*-ssDNA loop interaction might confer a conformation making adenine(s) in ssDNA loop more accessible to the active site of the latter TadA* as compared to the latter TadA*-ssDNA loop complex conformation. We further examined whether TadA* (LOF) could be replaced by TadA subunit and found that such replacement led to significant changes in editing activities or/and scopes (Fig. 2c, d). As it has been demonstrated that TadA* monomer fusing to nCas9 is sufficient for adenine deamination[12,15], we also generated ABE (TadA*)-nSpCas9-DS variants and assessed their activities against sgRNA-1. It was shown that most ABE (TadA*)-nSpCas9-DS variants retained the adenine deamination activities except for 583-/895-TadA* (Supplementary Fig. 3). Though the editing signatures were different from TadA* dimer counterparts, the editing efficiencies and scopes of functional ABE

(TadA*)-nSpCas9-DS variants were more similar to their corresponding ABE (TadA* (LOF)-TadA*)-nSpCas9-DS as compared to ABE (TadA*-TadA* (LOF))-nSpCas9-DS counterparts. These results indicated that, the TadA* monomer inside functional ABE (TadA*)-nSpCas9-DS variants binding to target genomic DNA regions might mainly serve as a scaffold for ssDNA loop, making adenine(s) accessible to TadA* in excessive free ABE (TadA*)-nSpCas9-DS proteins. Taken together, our analysis revealed that, in addition to the docking sites, inserted domains could also influence the editing signatures of ABE variants.

**Minimized RNA off-target activities of functional ABE-nSpCas9-DS variants.** RNA off-target activities hindered the wide applications of ABEs. We wondered whether domain insertion could reduce the RNA off-target activities. Functional ABE (TadA*-TadA*)-nSpCas9-DS variants (535-/583-/770-/793-/801-/895-/905-/919-1029-/1249-TadA*-TadA*), which displayed ≥40% A–G conversion efficiency, were all assessed by preliminary transcriptome analysis and it was revealed that domain insertion significantly reduced their RNA off-target activities as compared to N-TadA*-TadA* (Fig. 3a, b and Supplementary Fig. 4). It was noticed that domain insertion inside HNH domain led to the most significant reduction of RNA off-target activities. Indeed, the RNA off-target activity of 905-TadA*-TadA*, with insertion near the C-terminus of HNH domain, was almost eliminated. It is possible that TadA* dimer inserted inside HNH domain might be enwrapped, making the accession to RNAs more difficult.

As TadA* mutants that could reduce and even eliminate RNA off-target activities have been identified by deaminase engineering, we assessed whether RNA off-target activities could be further reduced by domain insertion in combined with TadA* mutants. Five ABE (TadA*-TadA*)-nSpCas9-DS variants (535-/770-/801-/1029-/1249-TadA*-TadA*), which displayed representative diversified editing scopes, and TadA* mutants including TadA* (K20A/R20A), TadA* (V106W), TadA* (V82G) and TadA* (F148A) were selected to generate ABE-nSpCas9-DS variants 535-/770-/801-/1029-/1249-TadA*(LOF)-TadA*(K20A/R20A)/-TadA*(V82G)/-TadA*(V106W)/-TadA*(F148A). As the former TadA* deaminase activity is unnecessary, TadA*(LOF) was used to reduce RNA off-target activities as demonstrated previously[17].

Firstly, systematic editing scope analysis was performed against four sgRNA sites for selected 535-/770-/801-/1029-/1249-TadA*-TadA* and derived 535-/770-/801-/1029-/1249-TadA*(LOF)-TadA*(K20A/R20A)/-TadA*(V82G)/-TadA*(V106W)/-TadA*(F148A) variants. 535-TadA*-TadA* displayed similar editing signatures to N-TadA*-TadA* and 1029-TadA*-TadA* displayed slightly expanded editing scopes with high editing activities at A2 for at least one sgRNA (Supplementary Fig. 5), while 770-/801-/1249-TadA*-TadA* displayed backward-shifted editing scopes, with maximal editing activities at A11-16 for 770-TadA*-TadA*, A11-14 for 801-/1249-TadA*-TadA* (Fig. 4a and Supplementary Fig. 5). For 535-/770-/801-/1029-/1249-TadA*(LOF)-TadA*(K20A/R20A)/-TadA*(V82G)/-TadA*(V106W)/-TadA*(F148A) variants, the maximal editing regions changed little while the editing scopes became more diversified as compared to their TadA*-TadA* counterparts (Fig. 4a, and Supplementary Figs. 6–9).

As TadA*(K20A/R20A) and TadA*(V82G) reduced RNA off-target activities similarly while TadA*(F148A) could eliminate RNA off-target activity, we chose N-/535-/770-/801-/1029-/1249-TadA* (LOF)-TadA*(V82G)/-TadA*(V106W) variants for RNA off-target analysis. In consistent with previous studies, the RNA off-target

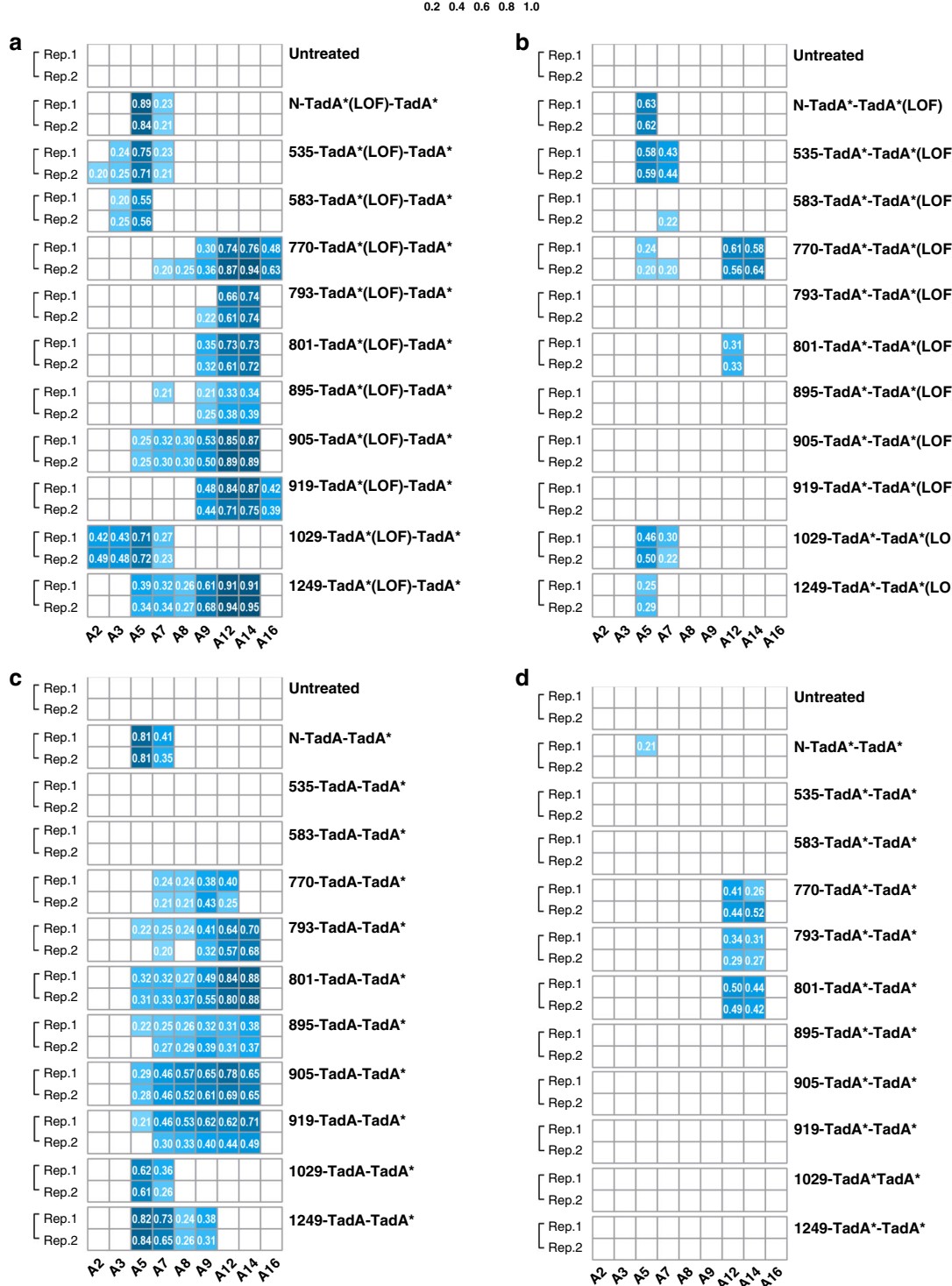

**Fig. 2 Base editing activities of functional ABE-nSpCas9-DS variants with different engineered TadA dimers.** Base editing activities of ABE (TadA* (LOF)-TadA*)-nSpCas9-DS variants (**a**), ABE (TadA*-TadA* (LOF))-nSpCas9-DS variants (**b**), ABE (TadA-TadA*)-nSpCas9-DS variants (**c**), and ABE (TadA*-TadA)-nSpCas9-DS variants (**d**) against sgRNA-1 were shown in heat map. Adenines in sgRNA-1 were labeled as red. A–G conversion frequencies at every adenine nucleotide in 20 bp protospacer were quantified with EditR. Two independent experiments were performed and editing frequencies >0.20 were labeled in the heat map. Source data are available in the Source data file.

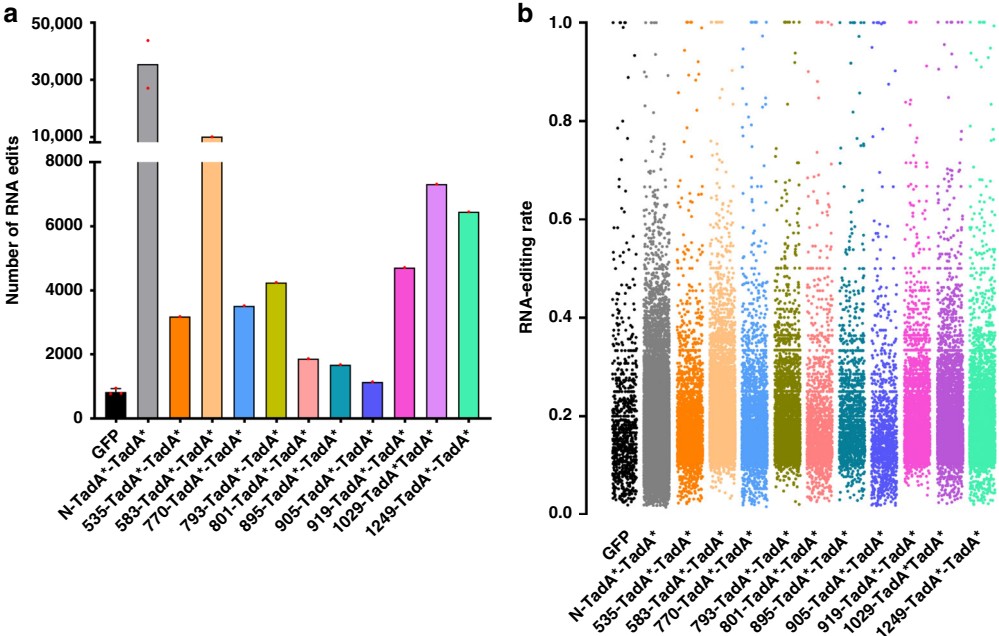

**Fig. 3 RNA off-target activities of functional ABE (TadA*-TadA*)-nSpCas9-DS variants and N-terminal counterpart. a** Transcriptome analysis showing the number of edited adenine nucleotides in HEK293T cells transfected with functional ABE (TadA*-TadA*)-nSpCas9-DS variants or N-terminal counterpart and sgRNA-1. (*n* = 3 biologically independent samples for GFP and data here are represented as mean with S.E.M., *n* = 2 biologically independent samples for N-TadA*-TadA* and *n* = 1 for other samples). **b** Representative jitter plots displaying the RNA A-to-I conversion frequencies at transcriptome level in HEK293T cells transfected with functional ABE (TadA*-TadA*)-nSpCas9-DS variants or N-terminal counterpart and sgRNA-1. Source data are available in the Source data file.

activities of N-TadA*(LOF)-TadA*(V82G)/-TadA*(V106W) were significantly reduced as compared to N- TadA*-TadA* but still induced about 22- and 7.5-fold higher numbers of edited adenines than background, respectively (Supplementary Fig. 10a, b).

Importantly, we found that the RNA off-target activities of 535-/770-/801-/1029-/1249-TadA*(LOF)-TadA*(V82G)/-TadA* (V106W) variants were further reduced as compared to their N-terminal counterparts (Fig. 4b and Supplementary Figs. 11 and 12). Among them, 1249-TadA*(LOF)-TadA*(V82G)/-TadA* (V106W) induced about 0.4-fold higher numbers while 1029-TadA*(LOF)-TadA*(V82G)/-TadA*(V106W) and 770-TadA* (LOF)-TadA*(V82G) induced about 1-fold higher numbers of edited adenines than background (Fig. 4b and Supplementary Figs. 11 and 12). On the other hand, the RNA off-target activities of 535-/801-TadA*(LOF)-TadA*(V82G)/-TadA*(V106W) and 770-TadA*(LOF)-TadA*(V106W) variants were completely eliminated (Fig. 4b and Supplementary Figs. 11 and 12).

**DNA off-target analysis and characterization of representative ABE-nSpCas9-DS variants.** We further evaluated the DNA off-target activities of all functional ABE (TadA*(LOF)-TadA*)-nSpCas9-DS variants using sgRNA-FAN (FANCF site) and sgRNA-HPRT (HPRT site) at two sites with well-defined DNA off-target sites by either GUIDE-seq or Digenome-seq platform (four potential off-target sites of sgRNA-FAN and two potential off-target sites of sgRNA-HPRT were evaluated)[22,23]. We found that for 535-/583-/1029-TadA*(LOF)-TadA* displaying maximal activities at A5, 583-TadA*(LOF)-TadA* displayed minimal editing activities at both on-target sites. 535-TadA*(LOF)-TadA* displayed comparable off-target effects at all six off-target sites as compared to N-terminal counterpart while higher off-target effects were observed at sgRNA-HPRT off2 site for 1029-TadA* (LOF)-TadA* (Supplementary Figs. 13 and 14). For 770-/793-/801-/895-/905-/915-/1249-TadA*(LOF)-TadA* displaying max-imal activities at A12–14, off-target effects were higher at several

sites as these ABE variants displayed expanded editing scopes as compared to N-/535-/583-/1029-TadA*(LOF)-TadA* variants (Supplementary Figs. 13 and 14). Nevertheless, 770-/793-/801-TadA*(LOF)-TadA* retained high specificity at both on-target sites, with the maximal off-target A–G frequency <50% of on-target A–G frequency.

As ABE-nSpCas9-DS-770 and -801 variants displayed repre-sentative shifted editing scopes different from existing ABE tools, and their RNA off-target activities were further reduced by introducing deaminase mutations such as V82G or V106W, we chose 770- and 801-TadA*(LOF)-TadA*(V82G) variants for editing analysis against another 20 genomic sites to characterize their editing signatures in diverse contexts, which would be valuable for their future applications. It was shown that 770-TadA*(LOF)-TadA*(V82G) displayed quite efficient A–G con-version activity, with an average editing activity >40% at positions 9–16 (Fig. 5a). For 801-TadA*(LOF)-TadA*(V82G), the editing signature is more complicated as compared to 770-counterpart, and each position at 9–15 contained at least one adenine (A) site with >40% editing efficiency while the editing activities at the same position of different genomic sites are more scattered (Fig. 5b). In addition to adenine positions, it was revealed that the editing efficiency could also be modulated by sequence context. Results showed that 770-TadA*(LOF)-TadA*(V82G) preferred TAG > CA > AA/GA (underline for target) (Fig. 5c). 801-TadA* (LOF)-TadA*(V82G) preferred TA/CA than AA/GA (Fig. 5d), but their preference is more obvious, which might account for the more scattered editing efficiencies at the same position of different genomic sites. In addition, it was shown that 801-TadA*(LOF)-TadA*(V82G) displayed slight preference for AG/AT than AA (Fig. 5d).

As shown above, we noticed that 1029-TadA*-TadA* edited A2 in sgRNA-1 and 770-TadA*-TadA* edited A16 with an efficiency >30%, which indicated that functional ABE (TadA*-TadA*)-nSpCas9-DS variants generated in this study could

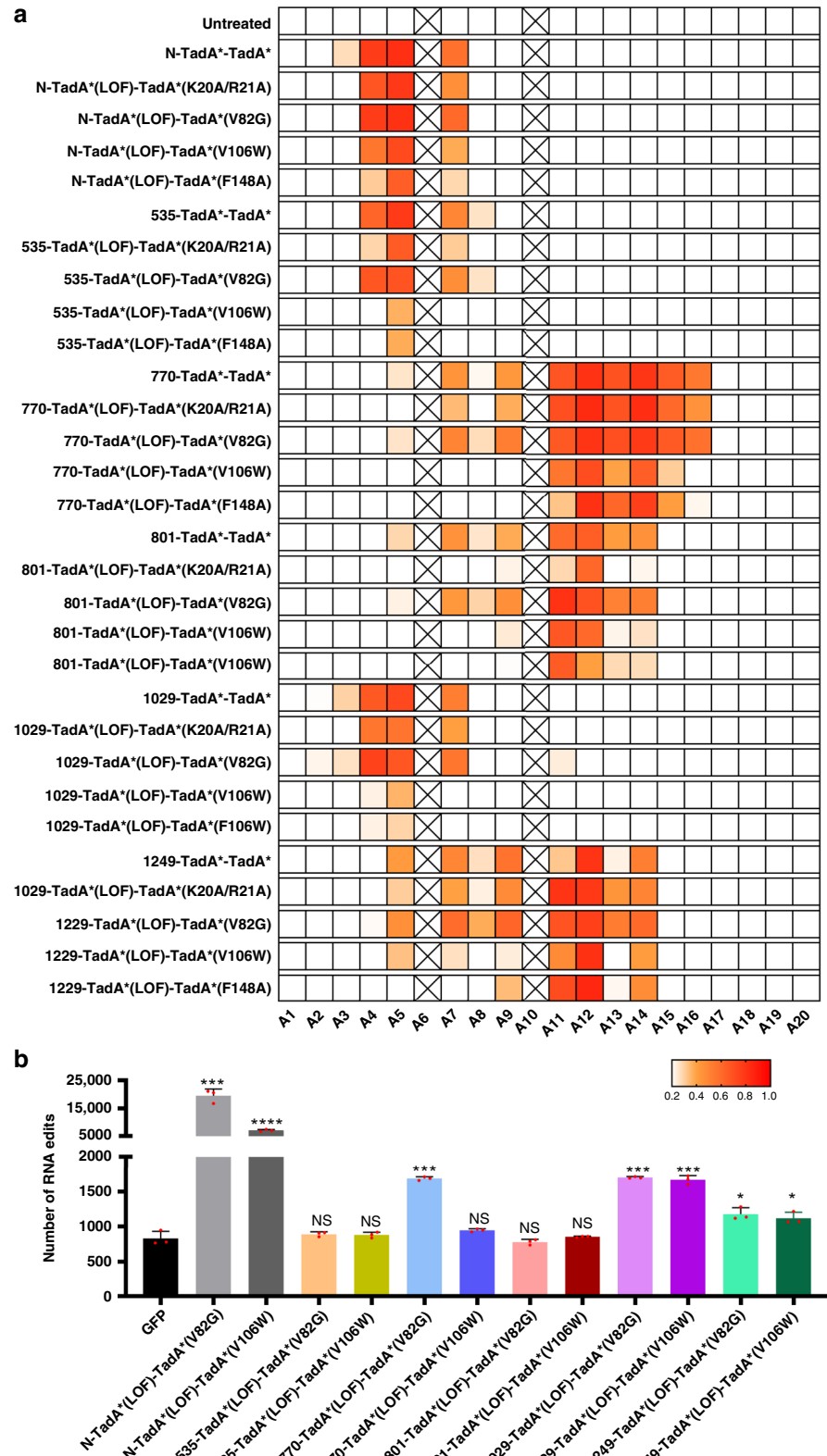

**Fig. 4 Comparison of editing scopes and RNA off-target activities for selected ABE-nSpCas9-DS variants. a** Base editing activities at each adenine position across four sgRNA sites were averaged and shown in heat map for selected ABE-nSpCas9-DS variants. Boxes with × are positions with no adenine nucleotide at all four sgRNA sites. **b** Transcriptome analysis showing the number of edited adenine nucleotides in HEK293T cells transfected with selected ABE-nSpCas9-DS variants or N-terminal counterparts and sgRNA-1. Data here are represented as mean with S.E.M. from three biologically independent experiments. Statistical analysis was performed for all experimental groups against GFP group and *$p < 0.05$, ***$p < 0.001$; ****$p < 0.0001$ with two-tailed unpaired t-test. n.s. means no significance. (from left to right, $p = 0.0002$; $p < 0.0001$; $p = 0.3825$; $p = 0.4781$; $p = 0.0001$; $p = 0.1230$; $p = 0.4492$; $p = 0.7062$; $p = 0.0001$; $p = 0.002$; $p = 0.0110$; $p = 0.0192$;) Source data are available in the Source data file.

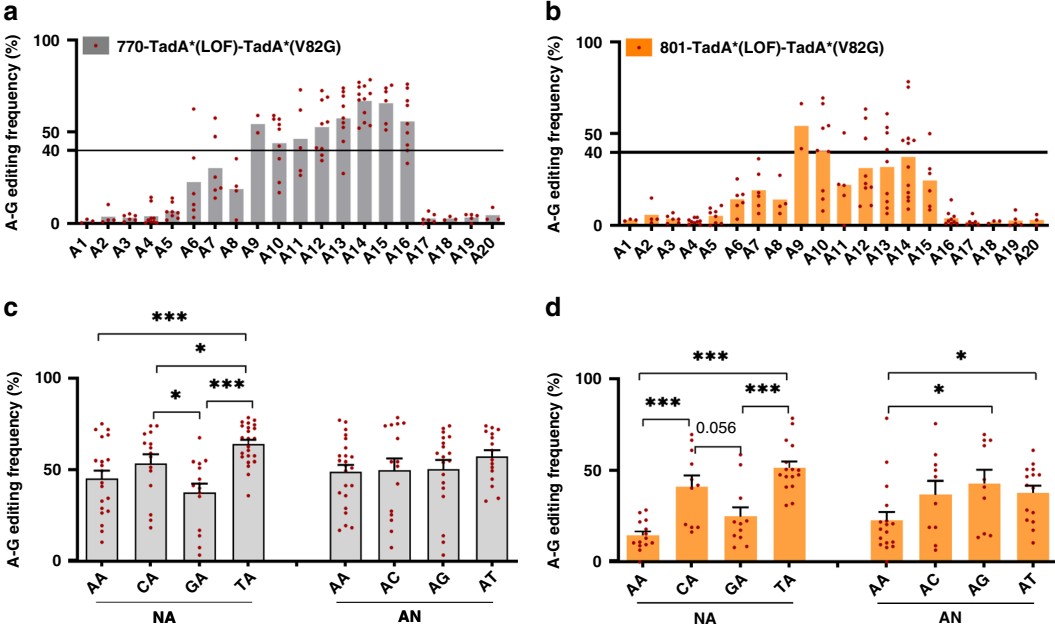

**Fig. 5 Comprehensive analysis of editing windows and sequence preference for 770- and 801-TadA* (LOF)-TadA* (V82G). a, b** A–G conversion frequencies at each adenine position across 20 sgRNA sites were quantified and summarized for 770- TadA* (LOF)-TadA* (V82G) (**a**) and 801-TadA* (LOF)-TadA* (V82G) (**b**). Dot plots in the figure are averages of $n = 2$ biologically independent experiments. **c, d** Sequence preference was analyzed for 770- TadA* (LOF)-TadA* (V82G) (**c**) and 801-TadA* (LOF)-TadA* (V82G) (**d**). *$p < 0.05$, ***$p < 0.001$; ****$p < 0.0001$ with two-tailed unpaired t-test. Dot plots in the figure are averages of $n = 2$ biologically independent experiments. Data here are represented as mean with S.E.M. of all dot plots in the same column. (For 770- TadA* (LOF)-TadA* (V82G), AA vs TA, $p = 0.0004$; CA vs GA, $p = 0.0287$; CA vs TA, $p = 0.0037$; GA vs TA, $p < 0.0001$. For 801-TadA* (LOF)-TadA* (V82G), AA vs CA, $p = 0.0002$; AA vs TA, $p < 0.0001$; CA vs GA, $p = 0.0056$; GA vs TA, $p = 0.0001$; AA vs AG, $p = 0.0217$; AA vs AT, $p = 0.0127$.) Source data are available in the Source data file.

achieve efficient A–G conversions across protospacer positions 2–16. Such broadened editing scopes would make more genomic sites targetable as compared to previous ABE tools with editing scopes across positions 4-8 or 4–14[6]. Indeed, Analysis of human pathogenic single-nucleotide polymorphisms (SNPs) in Clinvar database[4] revealed that the percentage of correctable pathogenic SNPs by ABE tools with editing scopes across 4–8, 4–14, and 2–16 was 30%, 51%, and 61%, respectively (Supplementary Fig. 15). Overall, functional ABE-nSpCas9-DS variants developed in this study expanded the A–G base editing toolkit and would further promote their applications in more research areas.

**Diversified cytosine deamination signatures of functional ABE-nSpCas9-DS variants.** Additionally, we noticed that N-TadA*-TadA* induced about 3% of C6 conversion at sgRNA-B site, in consistent with previous studies showing that ABEs could induce cytosine deamination at specific sites[15,20] (Fig. 6a). Similarly, 535-TadA*-TadA* induced >1% conversion efficiency for C5-C7 while 770-/801-/1249-TadA*-TadA* variants induced >1% conversion efficiency for C10-C13 at sgRNA-B site. Furthermore, obvious C11 conversions at sgRNA-16 site were also observed for 770-/801-/1249-TadA*-TadA* variants (Fig. 6a). Cytosine conversions were also assessed for representative ABE-nSpCas9-DS variants containing different TadA* mutants including TadA*(V82G), TadA*(F148A), TadA*(K20A/R21A) or TadA*(V106W) (Fig. 6b, c and Supplementary Fig. 16a, b). It was revealed that for N-terminal based ABEs, F148A and V82G increased the C6 conversion efficiency to ~4% and 8%, respectively (Fig. 6b, c) and V106W reduced the C6 conversion efficiency to ~1% (Supplementary Fig. 16a), while K20A/R21A had minor impact on C6 conversion efficiency at sgRNA-B site (Supplementary Fig. 16b), indicating a TadA* mutation-

dependent manner for cytosine conversion. Similarly, ABE-nSpCas9-DS535 variants induced >1% conversion efficiency for C5-C7 in a TadA* mutation-dependent manner (Fig. 6b, c and Supplementary Fig. 16a, b). We also found that ABE-nSpCas9-DS770/-DS801/-DS1249 variants mainly induced >1% conversion efficiency for C8-C13 in a TadA* mutation-dependent manner, with the highest up to ~41% at C8 for 1249- TadA*(LOF)-TadA* (V82G) (Fig. 6b). ABE-nCas9-DS770/-DS801/-DS1249 variants also induced obvious C11 conversion at sgRNA-16 site, with the highest up to ~22.8% for 1249-TadA*(LOF)-TadA*(V82G) (Fig. 6b). We also noticed that ABE-nCas9-DS1029 variants generally displayed reduced cytosine editing activity as compared to other variants, and the most obvious reduction was observed in 1029-TadA*(LOF)-TadA*(F148A) variant (Fig. 6a–c and Supplementary Fig. 16a, b). This might be attributed to the surrounding structural conformation of DS1029, which limited the accessibility of cytosine in ssDNA loop, and F148A further limited the cytosine deamination window of TadA*, as it has been reported that F148A narrowed the adenine deamination window[16]. Generally, it is possible that different DS sites for TadA* dimer insertion confer distinct structural conformation to access cytosine(s) in ssDNA loop, while deaminase engineering would change the cytosine deamination activities, with V82G enhancing while V106W inhibiting their cytosine deamination activities generally.

It was noted that our results are not entirely consistent with previous results, which showed that ABE(N-TadA-TadA*)-mediated cytosine conversion preferred TCN motif and a narrow editing window (positions 5–7)[20] while our ABE-nSpCas9-DS variants could also target C<u>C</u> (underline indicating the target site) in addition to TC with an expanded editing window (positions 5–13). These differences might be attributed to the following reasons. Firstly, TadA* are quite different from TadA in DNA

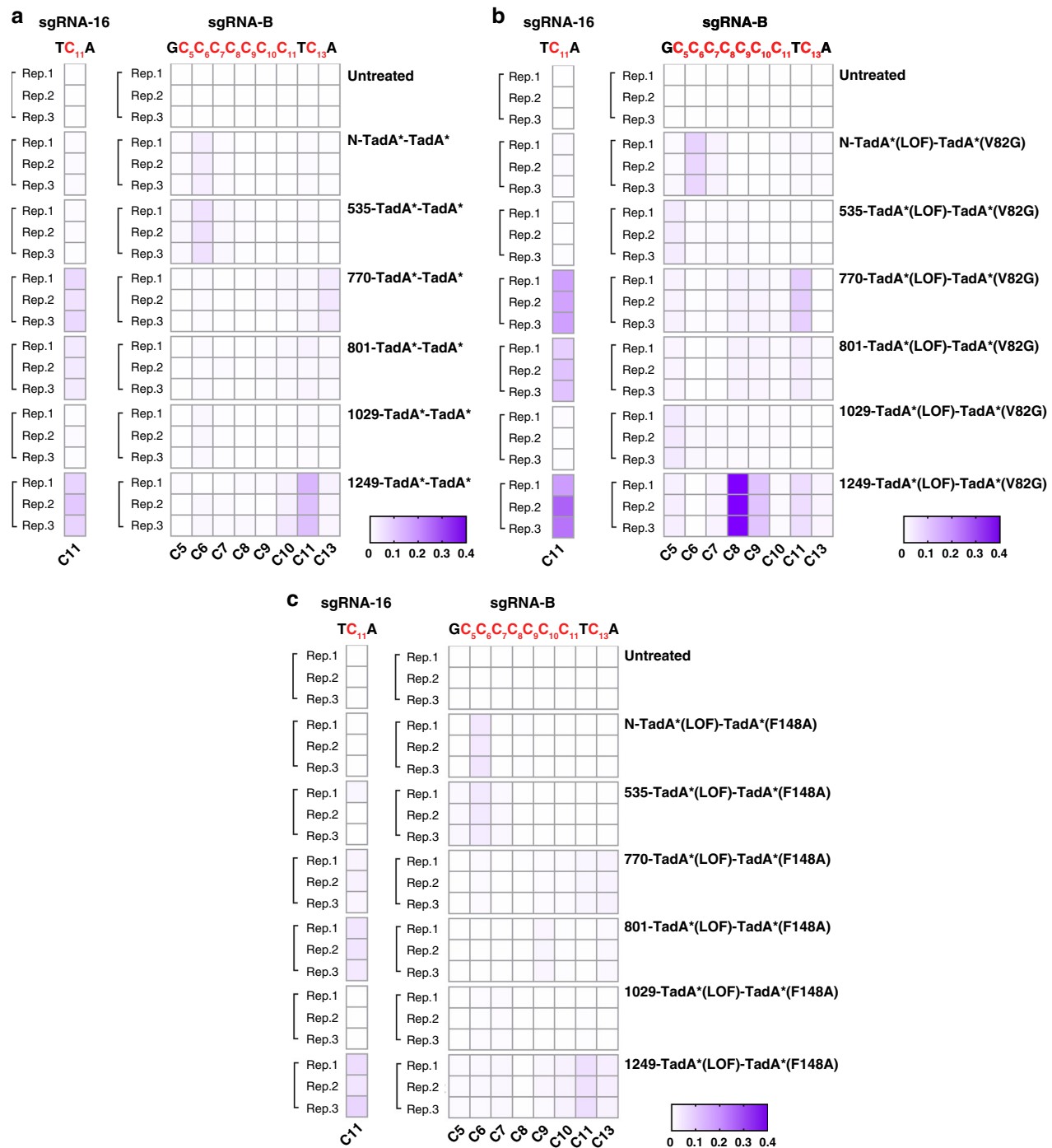

**Fig. 6 Cytosine deamination activities of selected ABE-nSpCas9-DS variants and N-terminal counterpart.** Cytosine deamination activities of selected ABE (TadA*-TadA*)-nSpCas9-DSs (**a**), ABE (TadA*(LOF)-TadA*(V82G))-nSpCas9-DSs (**b**), ABE (TadA*(LOF)-TadA*(F148A))-nSpCas9-DSs (**c**), and N-terminal counterpart against sgRNA-B and sgRNA-16. cytosines in sgRNAs were labeled as red. Source data are available in the Source data file.

binding and deamination abilities, which might lead to the changes of preferred target motif. In addition, TadA* dimer insertions at different DSs changed their relative distances to ssDNA loop and targetable cytosine(s), and thus resulted in editing window expansions. Systematic analysis would be needed to clarify the impact of DS locations in combined with deaminase engineering on ABE cytosine deamination activities, which might interfere with canonical adenine deamination activity, to provide guidance for the selection of ABE variants. ABE cytosine deamination activity also deserves further explorations to assess their applicability for cytosine conversions.

## Discussion

In this work, we generated a series of functional ABE variants with diversified editing scopes and minimized RNA off-target activities using domain insertion profiling, which confirmed that domain insertion is an effective and powerful strategy for the development of Cas9-based fusion proteins. When combined with other strategies such as deaminase engineering, the editing scopes and RNA off-target activities of ABE tools were further improved, which allow efficient A–G conversions at currently untargetable sites and address a major safety concern that hindered the broad applications of ABEs. We also notice that the

editing signatures of ABE variants are dependent on both docking sites and inserted deaminases, indicating that ABE toolkits could be further diversified by different combinations of TadA* variants. As ABE variants could display distinct editing signatures with different TadA* combinations at the same DSs, for every functional domain to be assessed, it is necessary to screen all potential DSs comprehensively to avoid omitting functional ABE variants. In addition to SpCas9-DSs, recent studies have established Cas9-CPs for fusion protein construction[11]. Both strategies are powerful for Cas9-based protein engineering. Indeed, the relative position of fusion domains could be changed when they were linked to either SpCas9-DSs or Cas9-CPs as compared to wild-type Cas9. For Cas9-CPs-based fusion proteins, they were still constructed through conventional "head-to-tail" fusion strategy, and thus the fusion domain is quite flexible as only one side is linked to Cas9-CPs. On the other hand, for SpCas9-DSs-based fusion proteins, domains were inside SpCas9-DSs, which would limit their excessive flexibility. Domain flexibility differences between Cas9-CPs- and SpCas9-DSs-based fusion proteins would affect protein properties such as protein stability, activity and specificity. As both strategies have been successfully applied for ABE engineering, we noticed that ABE (TadA-TadA*)-nSpCas9-DS1029/1249 displayed similar editing scopes as compared to CP1028/1249-ABEmax (TadA-TadA*) based on limited data, which indicated that similarity also existed between Cas9-CPs- and SpCas9-DSs-based tools[6]. Here we found that domain flexibility might be critical for RNA off-target effects, further analysis of RNA off-target differences between Cas9-CPs- and SpCas9-DSs-based ABE variants would be valuable to elucidate their characteristic differences. Additionally, Cas9-CPs without DNA cleavage activities but retaining DNA binding abilities remained to be discovered and evaluated in the future, as our results suggested that under specific conditions, DNA cleavage activity might be dispensable for SpCas9-DSs-based tools. Additionally, modest positive correlations have been observed between N-terminal-based conventional ABE/CBE activities and DNA cleavage activity of Cas9[24] while ABE-nSpCas9-DS770 to ABE-nSpCas9-DS919 described here maintained high editing activities without any DNA cleavage activity. As regions 770-919 mainly cover the HNH domain in Cas9, it is possible that deaminase inside here might induce specific conformation changes to promote base editing process. In summary, this study demonstrate that domain insertion provides a framework for further improvement of base editing tools and would generate more diversified and improved toolkits when combined with other strategies such as deaminase engineering. As different DSs inside Cas9 might confer distinct structural hindrance to modulate the functions of inserted domains, such divergence could be utilized to develop diversified tools such as ABEs with unique editing scopes and specificities to achieve more precise editing effects.

## Methods

**Cell culture and transfection**. HEK293T cells (Cell Bank of the Chinese Academy of Sciences (Shanghai, China)) were cultured in a 5% $CO_2$ incubator (Thermo Scientific Heraeus) using DMEM (Gibco/Life Technologies) containing 10% FBS (Gibco/Life Technologies) at 37 °C. For cell transfection, HEK293T cells were initially plated into 6-well or 24-well plates (Thermo Scientific) and cultured for 20 h. Then transfection was performed using Lipofectamine 3000 (Thermo Scientific) mixed with plasmid expressing specific base editor and plasmid expressing specific sgRNA (mole ratio 4:1). For base editing analysis, cells were cultured for 72 h for flow cytometry and genome isolation using lysis buffer (10 mM Tris-HCl, pH 8.8, 5 mM EDTA, 0.2% SDS, 0.2 M NaCl, 25 μg/mL Proteinase K (Calbiochem)). For transcriptome analysis, cells were cultured for 48 h for flow cytometry and RNA isolation using Trizol reagent (Life Technologies).

**Base editing analysis**. GFP/mCherry double-positive cells were isolated using a Moflo XDP (Beckman Coulter) and flow cytometry data was analyzed using

SUMMIT Version 5.2.0 software. Cells were sorted into 200 μL lysis buffer (10 mM Tris-HCl, pH 8.8, 5 mM EDTA, 0.2% SDS, 0.2 M NaCl, 25 μg/mL Proteinase K (Calbiochem)) for genome isolation. PCR primers were designed and synthesized (GENEWIZ). Amplification products were prepared using Takara LA Taq for sanger sequencing (Shanghai Majorbio Bio-Pharm Technology) and editing efficiency was quantified using EditR[21]. Sanger sequencing analysis was performed in biological duplicates and detailed Sanger sequencing data was summarized in Supplementary Data 2. Base editing efficiencies were also evaluated using high-throughput sequencing as described previously. In brief, PCR products with different barcodes were prepared using Universal DNA Purification Kit (Tiangen Biotech, Beijing, China). Library preparation and deep sequencing was performed using Illumina NovaSeq platform at NovelBio Bio-Pharm Technology, Shanghai, China. Base editing efficiencies were calculated as follows: FastQC (v0.11.4) was used to evaluate raw data and those with quality score < 15 were trimmed. Bowtie 2 (version 2.2.5) was used for data mapping and samtools (version 1.3.1) was used for substitution quantification. All samples were prepared in biological triplicates. Processed deep sequencing data was summarized in Supplementary Data 3. PCR primers were listed in Supplementary Data 4.

**Indel analysis for SpCas9-DS-ABEs**. Genome samples were prepared as described in base editing analysis. Amplification products were prepared using Takara LA Taq for sanger sequencing (Shanghai Personalbio Technology Co.,Ltd.) and Indel efficiency was quantified using online tool ICE analysis (https://ice.synthego.com/)[25].

**Sequence preference evaluation**. Sequence preference for 770- and 801-TadA* (LOF)-TadA*(V82G) was analyzed as described previously[10] with slight modifications[10]. Generally, positions containing at least one adenine (A) site with >40% editing efficiency were included for preference evaluation. A sites were classified according to NA or AN types. Statistical analysis was performed with two-tailed unpaired t-test and $p < 0.05$ was considered significant. Detailed Sanger sequencing data was summarized in Supplementary Data 2.

**Plasmid preparation**. Human codon optimized deaminases TadA and TadA* were synthesized commercially (Genescript). For mutagenesis, KOD-Plus system was used to generate TadA*(K20A/R21A), TadA*(V82G), TadA*(V106W), TadA*(F148A), TadA*(LOF), and docking sites inside SpCas9 (PX461 (Addgene #48140)) for fusing protein construction. For TadA* dimer insertions, docking site, and the N-terminus of TadA* dimer was connected via a linker (SARPKKKRK-VAAAGSGPKKKRKVAAAGSS) containing two NLS signals (underline). For TadA* monomer insertions, a GSS linker was used between the docking site and the N-terminus of TadA* monomer. The docking site was linked to the C-terminus of both TadA* dimer and monomer via ART amino acids. Plasmids for sgRNA expression were generated as described previously[10]. In brief, U6-sgRNA-EF1apha-UGI-T2A-mCherry plasmid was linearized with BsaI, and paired oligonucleotides were synthesized, annealed, and inserted for sgRNA expression vector constructions. All sgRNA information was listed in Supplementary Table 1. All primers for mutagenesis were listed in Supplementary Data 1.

**Analysis of RNA off-target activities with RNA sequencing**. For transcriptome analysis, base editors were co-transfected with sgRNA-1 for 48 h. And GFP/mCherry double-positive cells were isolated using a Moflo XDP (Beckman Coulter) and total RNA was isolated using Trizol reagent for RNA-seq. Illumina TruSeq RNA Sample Prep Kit (Cat#FC-122-1001) was used with 1 ug of total RNA for the construction of sequencing libraries with standard Illumina protocols. RNA-seq was performed using Illumina HiSeq X Ten platform in Shanghai Novelbio Ltd. For each library, ~50–60 million reads were generated. We applied the reads filtering towards the raw reads after sequencing to achieve the clean data following the criteria: a) 10% base quality <15 b) 13% base quality <20. RNA sequencing data was aligned to the human reference genome (GRCh38) using STAR (v2.5.2b). Variants were called using the GATK best practices pipeline using Picard and GATK 3.8. Single-nucleotide variants (SNVs) were filtered to include loci with reads >10 and labeled as A–G or T–C for to evaluate RNA off-target activities.

**Statistical analysis**. Statistical details are provided in the text, figure legends, and methods. Error bars were standard error of the mean (S.E.M.) from independent experiments/samples. Two-tailed unpaired Student's t-test was chosen for comparisons and statistical significance was considered as p value < 0.05. GraphPad Prism was used for statistical analysis of this study.

**Reporting summary**. Further information on research design is available in the Nature Research Reporting Summary linked to this article.

## Data availability

All raw high-throughput sequencing data are available in GSE142840 and BioProject: PRJNA598461. Source data are provided with this paper.

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

## Acknowledgements

We thank Yuefang Zhang, Shifang Shan and Ran Zhang for their help in cell culture and genome isolation. Haiyang Wu, Lijuan Quan, Songlin Qian and Min Zhang for the help in flow cytometry. This work was supported by grants from National Key R&D Program of China (2019YFA0111000), Natural Science Foundation of Shanghai (20ZR1403100), Shanghai Sailing Program (15YF1414200) to T.L.C., National Natural Science Foundation of China (#31600826) to T.L.C., and (#31625013, #81941405) to Z.Q., and (61932008, 61772368) to X.M.Z., The Strategic Priority Research Program of Chinese Academy of Science, Grant No. XDBS01060200, the Shanghai Municipal Science and Technology Major Project (#2018SHZDZX05) and the Shanghai Brain-Intelligence Project from STCSM (16JC1420501) to Z.Q., The Rong-Chang Charity Fund of Shanghai Charity Foundation and Zhongshan Hospital to X.L.W., Shanghai Municipal Science and Technology Major Project (2018SHZDZX01) and ZJLab to X.M.Z. The research is supported by the Open Large Infrastructure Research of Chinese Academy of Sciences.

## Author contributions

T.L.C. designed the study. T.L.C., S.L., J.L.C., and J.Y.Q. performed experiments; J.X.C., J.Q.C., and B.Y. performed computational analysis. T.L.C., X.L.W., Z.Q. and X.M.Z. supervised the study. All authors contributed to writing the paper.

## Competing interests

The authors declare the following competing financial interests. Zhongshan Hospital, Fudan University has a patent (202011117935.X) pending, with S.L. and X.L.W. listed as inventors, concerning the functional ABE variants described in this paper. The remaining authors declare no competing interests.
