## [Peer Review File · Nature Communications]

Reviewers' Comments:

Reviewer #1:

Remarks to the Author:

Li et al presents "Docking sites inside Cas9 for adenine base editing diversification and RN2 off-target elimination", a manuscript that provides an engineering approach and expanded adenine base editor (ABE) tools. As a theoretical ~50% of disease-associated single-nucleotide polymorphisms could be reverted by ABEs, there is tremendous interest in new and improved approaches, particularly in expanding the positional constraints of the modified A's inside the gRNA sequence. Here, the authors reason that Cas9 can be modified with domain insertion (as has been previously shown) to expand the scope of the ABE editing. Further, they provide proof-of-principle results to further reduce off-targeting RNA-editing in their new editors. I believe the work to be of generally high quality and warrants publication, noting a few suggestions.

Comments

- 1) I generally disagree with the presentation of the editing activity in throughout the manuscript. In the figures, the authors report the mean over 2 experiments and abstract from the actual editing efficiencies by showing a relative metric in the heatmaps or by arbitrarily defining cutoffs in Table 1. I would suggest that the authors portray their data as has been more common in other base editing manuscripts from the Liu and/or Joung labs. Specifically, reporting results in units of nucleotides edited and stacking replicates on the heatmap would be beneficial for transparency and be more consistent with the existing literature of base editing
- 2) The authors do not provide any sort of off-target analysis. Even if a complete unbiased search for off target activity is impractical, examining the top predicted off target sites is essential for evaluating the efficacy of these new base editors.
- 3) Reporting performance of the top performing engineered variants on additional genomic loci would be an important addition for readers to understand the efficacy of the improved enzymes in diverse contexts.
- 4) To better contextualize the advances, a stronger tie to disease relevance could be beneficial. If not with another application to a disease-associated locus (in line with comment #3), could the authors provide computational analyses to identify the additional number (or percent) of disease-associated or pathogenic variants (perhaps from ClinVar) that could be 'corrected' using their new variants?

Reviewer #2:

Remarks to the Author:

General comments;

DNA base editing technologies, including cytosine base editors (CBEs) and adenine base editors (ABEs), have undoubtedly revolutionized the genome editing field. However, several studies have raised drawbacks such as limited target window, DNA/RNA off-target mutations, and ABE cytosine deamination activity. In this study, the authors prepared SpCas9-DS variants containing various docking sites for adenosine deaminase conjugation. Indeed, lots of experiments with many different SpCas9-DS variants and different Tada* mutants, the authors constructed significantly meaningful ABE variants that have advantages such as broaden target window, reduced RNA off-target effect, and reduced ABE-mediated cytosine conversion. Overall, this manuscript is straightforward and the topic of this manuscript is potentially suitable for this journal, Nature Communications, but I raise a few issues that can strengthen this study.

Specific comments;

I feel that the interpretation of experimental results and discussion are poor.

1) Abstract of current version has very limited information. For example, it would be better to describe more about the detailed information such as final recommended feature of improved ABE.

2) In the scheme of SpCas9-DS of figure 1a, the reason why to choose 24 docking sites inside SpCas9 should be described for reader's understanding. And could the author confirm that the natural SpCas9 activity was maintained after inserting DSs there? In other ward, If one select another sites for DS insertion, the SpCas9's activity may be lost?

3) In figure 1c, I am curious whether the ABE (TadA*-TadA*)-nSpCas9-DS variants can target A18-20 in addition to A16, when the target DNA contain A18-20, because 770-ABE-DS and 919-ABE-DS still seem to have high activities for A16. It is necessary to test for another target sites having A18-20, which may suggest broad option of ABE targetable window from A1 to A20.

4) In figure 2, the experimental data are really interesting but the interpretation seems to be poor here. Why did the version of TadA*(LOF)-TadA* show different outcomes with that of TadA*-TadA*(LOF)? And why did the version of TadA* monomer (Figure S2) showed similar feature to that of TadA*(LOF)-TadA*? It should be discussed.

5) In figure 3, it is really interesting that 905-TadA*-TadA* showed highly reduced RNA off-target effects compared to canonical N-TadA*-TadA*. This data is meaningful because previous studies improved TadA itself, instead of SpCas9 shown in this study. The reason should be discussed more here.

6) In figure 5, the ABE-nSpCas9-DS variants cytosine conversion seems to be much different from the previous results (Ref 20), in which ABE cytosine conversion showed preferred target motif (TCN) and confined narrow target window. Furthermore, it is notable that 1029-TadA*(LOF)-TadA*(F148A) showed almost reduced cytosine editing. The reason should be discussed here more.

7) It is necessary to explain the functional and characteristic differences between SpCas9 circular permutants (Cas9-CPs) and SpCas9-DS at least in the discussion part.

8) The information of linker between TadA enzyme and Cas9 should be described in the methods part.

Response to reviewers' comments (Comments in black):

REVIEWER COMMENTS

Reviewer #1 (Remarks to the Author):

Li et al presents “Docking sites inside Cas9 for adenine base editing diversification and RN2 off-target elimination”, a manuscript that provides an engineering approach and expanded adenine base editor (ABE) tools. As a theoretical ~50% of disease-associated single-nucleotide polymorphisms could be reverted by ABEs, there is tremendous interest in new and improved approaches, particularly in expanding the positional constraints of the modified A’s inside the gRNA sequence. Here, the authors reason that Cas9 can be modified with domain insertion (as has been previously shown) to expand the scope of the ABE editing. Further, they provide proof-of-principle results to further reduce off-targeting RNA-editing in their new editors. I believe the work to be of generally high quality and warrants publication, noting a few suggestions.

Comments

1) I generally disagree with the presentation of the editing activity in throughout the manuscript. In the figures, the authors report the mean over 2 experiments and abstract from the actual editing efficiencies by showing a relative metric in the heatmaps or by arbitrarily defining cutoffs in Table 1. I would suggest that the authors portray their data as has been more common in other base editing manuscripts from the Liu and/or Joung labs. Specifically, reporting results in units of nucleotides edited and stacking replicates on the heatmap would be beneficial for transparency and be more consistent with the existing literature of base editing

Responses: Thanks a lot for the reviewer’s suggestions, and we have re-organized the data in the revised manuscript for Fig.1c, Fig.2, Fig.6 and Sup.Fig.3. Additionally, the activity description in Table 1 was deleted.

2) The authors do not provide any sort of off-target analysis. Even if a complete unbiased search for off target activity is impractical, examining the top predicted off target sites is essential for evaluating the efficacy of these new base editors.

Responses: Thanks a lot for the reviewer’s suggestions, and in the revised manuscript, the top predicted off target sites for two genome locus (FANCF, HPRT) were analyzed and described as follows: “We further evaluated the DNA off-target activity of all functional ABE (TadA*(LOF)-TadA*)-nSpCas9-DS variants using sgRNA-FAN (FANCF site) and sgRNA-HPRT (HPRT site) at two sites with well-defined DNA off-target sites by either GUIDE-seq or Digenome-seq platform (four potential off-target sites of sgRNA-FAN and two potential off-target sites of sgRNA-HPRT were evaluated)^{22, 23}. We found that for 535-/583-/1029-TadA*(LOF)-TadA* displaying maximal activities at A5, in addition to 583-TadA*(LOF)-TadA* displaying minimal editing activities at both on-target sites, 535-TadA*(LOF)-TadA* displayed comparable off-target effects at all six off-target sites as compared to N-terminal counterpart while higher off-target effects at were observed at sgRNA-HPRT off2 site for 1029-TadA*(LOF)-TadA*(Supplementary Fig. 13-14). For

770-/793-/801-/895-/905-/915-/1249-TadA*(LOF)-TadA* displaying maximal activities at A12-14, off-target effects were higher as these ABE variants displayed expanded editing scopes as compared to N-/535-/583-/1029-TadA*(LOF)-TadA* variants (Supplementary Fig. 13-14). Nevertheless, 770-/793-/801-TadA*(LOF)-TadA* retained high specificity at both on-target sites, with the maximal off-target A-G frequency less than 50% of on-target A-G frequency.”

3) Reporting performance of the top performing engineered variants on additional genomic loci would be an important addition for readers to understand the efficacy of the improved enzymes in diverse contexts.

Responses: Thanks a lot for the reviewer’s suggestions, and in the revised manuscript, two representative engineered variants 770-TadA*(LOF)-TadA*(V82G) and 801-TadA*(LOF)-TadA*(V82G), which display shifted editing scopes not reported previously, were evaluated on 20 additional genomic loci as follows: “As ABE-nSpCas9-DS-770 and -801 variants displayed representative shifted editing scopes different from existing ABE tools, and their RNA off-target activities were further reduced by inducing deaminase mutations such as V82G or V106W, we chose 770- and 801-TadA*(LOF)-TadA*(V82G) variants for editing analysis on another 20 genomic sites to characterize their editing signatures in diverse contexts, which would be valuable for their future applications. It was shown that 770-TadA*(LOF)-TadA*(V82G) displayed quite efficient A-G conversion activity, with an average editing activity >40% at positions 9-16 (Fig. 5a). For 801-TadA*(LOF)-TadA*(V82G), the editing signature is more complicated as compared to 770-counterpart, and positions at 9-15 contained at least one adenine (A) site with > 40% editing efficiency while the editing activities at the same position of different genomic sites are more scattered (Fig. 5b). In addition to adenine positions, it was revealed that the editing efficiency could also be modulated by sequence context. Results showed that 770-TadA*(LOF)-TadA*(V82G) preferred TA>CA>AA/GA (underline for target) (Fig. 5c). 801-TadA*(LOF)-TadA*(V82G) preferred TA/CA than AA/GA (underline for target) (Fig. 5d), but their preference is more obvious, which might account for the more scattered editing efficiencies at the same position of different genomic sites. Additionally, it was shown that 801-TadA*(LOF)-TadA*(V82G) displayed slight preference for AG/AT than AA (Fig. 5d).”

4) To better contextualize the advances, a stronger tie to disease relevance could be beneficial. If not with another application to a disease-associated locus (in line with comment #3), could the authors provide computational analyses to identify the additional number (or percent) of disease-associated or pathogenic variants (perhaps from ClinVar) that could be ‘corrected’ using their new variants?

Responses: Thanks a lot for the reviewer’s suggestions, and we have analyzed Clinvar database and found that our ABE-nSpCas9-DSs (editing windows from 2-16) could correct 61% pathogenic SNPs mutated from G to A or C to T, while previous reported tools could correct 30% or 51% with editing windows from 4-8 or 4-14, respectively. In the revised manuscript, it was described as follows: “As shown above, we noticed that 1029-TadA*-TadA* edited A2 in sgRNA-1 and 770-TadA*-TadA* edited A16 with an efficiency >30%, which indicated that functional ABE (TadA*-TadA*)-nSpCas9-DS variants generated in this study could achieve efficient A-G conversions across protospacer positions 2-16. Such broadened editing scopes would make more genomic sites targetable as compared to previous ABE tools with editing scopes across positions

4-8 or 4-14. Indeed, Analysis of human pathogenic single nucleotide polymorphisms (SNPs) in Clinvar database revealed that the percentage of correctable pathogenic SNPs by ABE tools with editing scopes across 4-8, 4-14 and 2-16 was 30%, 51% and 61%, respectively (Supplementary Fig. 15). Overall, functional ABE-nSpCas9-DS variants developed in this study expanded the A-G base editing toolkit and would further promote their applications in more research areas.”

Reviewer #2 (Remarks to the Author):

General comments;

DNA base editing technologies, including cytosine base editors (CBEs) and adenine base editors (ABEs), have undoubtedly revolutionized the genome editing field. However, several studies have raised drawbacks such as limited target window, DNA/RNA off-target mutations, and ABE cytosine deamination activity. In this study, the authors prepared SpCas9-DS variants containing various docking sites for adenosine deaminase conjugation. Indeed, lots of experiments with many different SpCas9-DS variants and different TadA* mutants, the authors constructed significantly meaningful ABE variants that have advantages such as broaden target window, reduced RNA off-target effect, and reduced ABE-mediated cytosine conversion. Overall, this manuscript is straightforward and the topic of this manuscript is potentially suitable for this journal, Nature Communications, but I raise a few issues that can strengthen this study.

Specific comments;

I feel that the interpretation of experimental results and discussion are poor.

1) Abstract of current version has very limited information. For example, it would be better to describe more about the detailed information such as final recommended feature of improved ABE.
Responses: Thanks a lot for the reviewer’s suggestions, we have revised the abstract as follows, with modified sentences and words emphasized as bold and underline:

“Nevertheless, current SpCas9-based adenine base editors (ABEs) with minimized RNA off-target activities display constrained editing scopes **with efficient editing activities at positions 4-8**. Here, functional ABE variants with diversified editing scopes and reduced RNA off-target activities were identified using domain insertion profiling inside *streptococcus pyogenes* (SpCas9) and with different combinations of TadA variants. **Engineered ABE variants in this study display narrowed, expanded or shifted editing scopes with efficient editing activities across protospacer positions 2-16.”**

2) In the scheme of SpCas9-DS of figure 1a, the reason why to choose 24 docking sites inside SpCas9 should be described for reader’s understanding. And could the author confirm that the natural SpCas9 activity was maintained after inserting DSs there? In other ward, If one select another sites for DS insertion, the SpCas9’s activity may be lost?

Responses: Thanks a lot for the reviewer’s suggestions, and the selection criteria for 24 docking

sites was provided in supplementary discussion as follows: “It has been demonstrated that SpCas9 protein could be engineered in many different ways while maintaining its RNA-guided double-stranded DNA (dsDNA) binding and cleavage ability. For example, Cas9 could be split into two fragments to establish a functional split-Cas9 system for inducible genome modification¹⁻³. Additionally, hotspots inside Cas9 tolerating mouse alpha1-syntrophin (PDZ) domain insertions have been identified⁴ and Cas9 could be rearranged via protein circular permutation strategy⁵. We speculate that the locations of split sites, functional insertional hotspots and hotspots of functional Cas9 circular permutants represent the potential regions tolerating additional domain insertions. In consideration of reported functional regions, we initially selected potential docking sites (DSs) around regions suitable for at least 2 out of 3 Cas9 engineering strategies including split-Cas9 construction, PDZ domain insertion and Cas9 circular permutation. These regions include DS¹¹³, DS²⁰³, DS³¹², DS⁴⁵⁹, DS⁵³⁵, DS⁶⁸⁷, DS⁷¹⁵, DS⁸⁰¹, DS⁹⁴⁶, DS¹⁰¹⁰, DS¹⁰²⁹, DS¹¹¹⁷, DS¹¹⁵⁴, DS¹²⁴⁹ and DS¹²⁸². As site 573-574 and site 713-714 have been used widely for functional split-Cas9 system construction, we further selected DS⁵⁸³, DS⁷⁰¹ and DS⁷³⁰ to evaluate their tolerating ability for adenosine deaminase insertions. Additionally, as HNH domain is intrinsically flexible inside Cas9 and close to ssDNA loop⁶, we also selected DS⁷⁷⁰, DS⁷⁹³, DS⁸⁹⁵, DS⁹⁰⁵ and DS⁹¹⁹ to assess their tolerating ability for adenosine deaminase insertions. As DS¹³²⁹ is near the PAM region, we wondered whether adenosine deaminase in this region could achieve A-G conversion around A17-A20, which are untargetable with existing ABE tools. The final 24 potential docking sites (DSs) are distributed across different domains of Cas9 and mainly in discrete and flexible regions. Thus, these 24 DSs could provide a relatively systematic assessment for the impact of adenosine deaminase insertions in SpCas9. However, as SpCas9 contains 1368 amino acids, a more comprehensive analysis for other regions is still needed in the future.”

Additionally, we also analyzed the natural SpCas9 activity after inserting DSs there against two sgRNAs and results were shown in the revised manuscript as follows: “We wondered whether the insertion of Tada*-Tada* dimer domain would interfere with SpCas9’s DNA cleavage activity. ABE (Tada*-Tada*)-SpCas9-DS variants were generated and their DNA cleavage activities were assessed against two sgRNAs. It was revealed that insertions at different DSs have distinct impact on SpCas9’s DNA cleavage activity (Supplementary Fig. 2) and no obvious correlation was observed between the base editing activities of ABE (Tada*-Tada*)-nSpCas9-DS variants and the DNA cleavage abilities of corresponding ABE (Tada*-Tada*)-SpCas9-DS counterparts. For example, ABE (Tada*-Tada*)-nSpCas9-DS770-919, with insertions around and inside HNH domain, displayed robust A-G conversion activities, while their corresponding ABE (Tada*-Tada*)-SpCas9-DS770-919 variants lost DNA cleavage activities because of HNH nuclease disruption (Supplementary Fig. 2). Additionally, it was shown that several ABE (Tada*-Tada*)-SpCas9-DS variants showed site-dependent DNA cleavage activities. For example, ABE (Tada*-Tada*)-SpCas9-DS1029 and -DS1117 displayed comparable cleavage activities against sgRNA-1 while their activities against sgRNA-E8 decreased significantly as compared to wild-type SpCas9 (Supplementary Fig. 2).”

3) In figure 1c, I am curious whether the ABE (Tada*-Tada*)-nSpCas9-DS variants can target A18-20 in addition to A16, when the target DNA contain A18-20, because 770-ABE-DS and 919-ABE-DS still seem to have high activities for A16. It is necessary to test for another target

sites having A18-20, which may suggest broad option of ABE targetable window from A1 to A20.

Responses: Thanks a lot for the reviewer's suggestions. As shown in the figure 1c, 770-ABE-DS have the highest activities for A16, the activities of 770-ABE-DS variants have been further examined for another three sgRNAs including sgRNA-16 (containing A17), sgRNA-B (containing A18, A20) and sgRNA-EMX (containing A17,A19,A20) (Supplementary Figure.). Results showed that 770-ABE-DS variants could not target A17-A20. Additionally, we also evaluated the editing signatures of 770-TadA*(LOF)-TadA*(V82G) against another 20 genomic loci, and no obvious editing activity was observed at A17-A20 (Fig. 5a in the revised manuscript).

4) In figure 2, the experimental data are really interesting but the interpretation seems to be poor here. Why did the version of TadA*(LOF)-TadA* show different outcomes with that of TadA*-TadA*(LOF)? And why did the version of TadA* monomer (Figure S2) showed similar feature to that of TadA*(LOF)-TadA*? It should be discussed.

Responses: Thanks a lot for the reviewer's suggestions. We discussed the differences between TadA*(LOF)-TadA* and TadA*-TadA*(LOF) in the revised manuscript as follows: “As both TadA* monomers inside nSpCas9 could serve as scaffold for ssDNA loop, above results indicated that the former TadA*-ssDNA loop interaction might confer a conformation making adenine(s) in ssDNA loop more accessible to the active site of the latter TadA* as compared to the latter TadA*-ssDNA loop complex conformation.”

Additionally, the editing similarity between TadA* monomer and TadA*(LOF)-TadA* was also discussed in the revised manuscript as follows: “Though the editing signature was different from TadA* dimer counterparts, the editing efficiencies and scopes of functional ABE (TadA*)-nSpCas9-DS variants were more similar to their corresponding ABE (TadA*(LOF)-TadA*)-nSpCas9-DS as compared to ABE (TadA*-TadA*(LOF))-nSpCas9-DS counterparts. These results indicated that, the TadA* monomer inside functional ABE (TadA*)-nSpCas9-DS variants binding to target genomic DNA regions might mainly serve as a scaffold for ssDNA loop, making adenine(s) accessible to TadA* in excessive free ABE (TadA*)-nSpCas9-DS proteins.”

5) In figure 3, it is really interesting that 905-TadA*-TadA* showed highly reduced RNA off-target effects compared to canonical N-TadA*-TadA*. This data is meaningful because previous studies improved TadA itself, instead of SpCas9 shown in this study. The reason should be discussed more here.

Responses: Thanks a lot for the reviewer's suggestions. We discussed the highly reduced RNA off-target effects of 905-TadA*-TadA* as follows: “It was noticed that domain insertion inside HNH domain led to the most reduction of RNA off-target activities as compared to other domains. Indeed, the RNA off-target activity of 905-TadA*-TadA*, with insertion near the C-terminus of HNH domain, was almost eliminated. It is possible that TadA* dimer inserted inside HNH domain might be enwrapped, making it inaccessible to RNAs and thus no obvious RNA off-target activities could be observed.”

6) In figure 5, the ABE-nSpCas9-DS variants cytosine conversion seems to be much different from the previous results (Ref 20), in which ABE cytosine conversion showed preferred target motif (TCN) and confined narrow target window. Furthermore, it is notable that

1029-TadA*(LOF)-TadA*(F148A) showed almost reduced cytosine editing. The reason should be discussed here more.

Responses: Thanks a lot for the reviewer's suggestions. The cytosine conversion difference between ABE-nSpCas9-DS variants and previous results was discussed in the revised manuscript as follows: "It was noted that our results are not entirely consistent with previous results, which showed that ABE(N-TadA-TadA*)-mediated cytosine conversion showed preferred target motif (TCN) and confined narrow target window (5-7)²⁰ while our ABE-nSpCas9-DS variants could also target CC (underline indicating the target site) in addition to TC and the target window even cover C5-C13. These differences might be attributed to the following reasons. Firstly, the characteristics of TadA* are quite different from TadA in DNA binding and deamination abilities, which might lead to the changes of preferred target motif. In addition, TadA* dimer insertions at different DSs changed their relative distances to ssDNA loop and targetable cytosine(s), and thus resulted in the changes of the target window."

Additionally, we also discussed the possible reason for the significant reduced cytosine editing activity of 1029-TadA*(LOF)-TadA*(F148A) as follows: "We also noticed that ABE-nCas9-DS1029 variants generally displayed reduced cytosine editing activity as compared to other variants, and the most obvious reduction was observed in 1029-TadA*(LOF)-TadA*(F148A) variant (Fig. 6a-c, Supplementary Fig. 16a-b). This might be attributed to the surrounding conformation of DS1029, which limited the accessibility of cytosine in ssDNA loop, and F148A further limited the cytosine deamination window of TadA*, as it has been reported that F148A narrowed the adenine deamination window¹⁶."

7) It is necessary to explain the functional and characteristic differences between SpCas9 circular permutants (Cas9-CPs) and SpCas9-DS at least in the discussion part.

Responses: Thanks a lot for the reviewer's suggestions. The functional and characteristic differences between SpCas9 circular permutants (Cas9-CPs) and SpCas9-DSs were discussed in the revised manuscript as follows: "In addition to SpCas9-DSs for the generation of functional fusion proteins via domain insertion strategy, recent studies have established Cas9-CPs for fusion protein construction¹¹. Both strategies are powerful for Cas9-based protein engineering. Indeed, the relative position of fusion domain could be changed when they were linked to either SpCas9-DSs or Cas9-CPs as compared to wild-type Cas9. For Cas9-CPs-based fusion proteins, they were still constructed through conventional "head-to-tail" fusion strategy by fusing specific domains to N- or C-terminus, and thus the fusion domain would be quite flexible as only one side is linked to Cas9-CPs. On the other hand, for SpCas9-DSs-based fusion proteins, domains were inserted inside SpCas9-DSs, which would limit the excessive flexibility of fusion domains. The different domain flexibility between Cas9-CPs- and SpCas9-DSs-based fusion proteins would affect protein properties such as protein stability, activity and specificity. As both strategies have been successfully applied for ABE engineering, we noticed that ABE (TadA-TadA*)-nSpCas9-DS1029/1249 displayed similar editing scopes as compared to CP1028/1249-ABEmax(TadA-TadA*) based on limited data, indicating that similarity also existed between Cas9-CPs- and SpCas9-DSs-based tools⁶. As domain flexibility is critical for RNA off-target effects shown in our study, further analysis of RNA off-target differences between Cas9-CPs- and SpCas9-DSs-based ABE variants would be valuable to elucidate their characteristic differences. Additionally, as functional Cas9-CPs were screened based on their

DNA cleavage activities, while our results suggested that under specific conditions, DNA cleavage activity might be dispensable for SpCas9-DSs-based tools, Cas9-CPs without DNA cleavage activities while maintaining DNA binding abilities remained to be discovered and evaluated in the future.”

8) The information of linker between TadA enzyme and Cas9 should be described in the methods part.

Responses: Thanks a lot for the reviewer’s suggestions. The information of the linker was added in the revised manuscript as follows: “For TadA* dimer insertions, docking site and the N-terminus of TadA* dimer was connected via a linker (SARPKKKRKVAAAGSGPKKKRKVAAAGSS) containing two NLS signals (underline). For TadA* monomer insertions, a GSS linker was used between docking site and the N-terminus of TadA* monomer. The docking site was linked to the C-terminus of both TadA* dimer and monomer via ART amino acids.”

Reviewers' Comments:

Reviewer #1:

Remarks to the Author:

The authors have made an exceptional effort to include new data in the manuscript, which I believe provides greater context for the efficacy of their new editors in diverse genomic contexts.

My only remaining (minor) comments involve the interpretability of the color bars in the heatmaps across figures 1,2, and 4 (which appear to still be the relative 'activity' measure) compared to Figure 6, which shows the allele frequency.

Minimally, the authors should provide labels in the figures and potentially the legends to describe these values. That said, providing all data in units of the observed allele frequencies seems the most intellectually honest for all panels and would be more in line with prior work in this space. I'd recommend the authors update these unless they have a good reason to continue utilizing the current presentation.

Reviewer #2:

Remarks to the Author:

The authors have mostly answered the issues I raised in the earlier review. At this time, I strongly feel that the strategy of using SpCas9-DS variants largely affect both Cas9 nuclease and adenosine deaminase. For example, the SpCas9-mediated DNA cleavage activities (Supplementary Figure 2) are not correlated (rather, negatively correlated) with the ABE-mediated base editing activities. i.e. ABEs from 770-TadA*-TadA* to 919-TadA*-TadA* did not generate indel mutations but were very effective for base editing. The canonical SpCas9-mediated DNA cleavage activities are positively correlated with the ABE/CBE-mediated base editing activities [Figure 1d at Nature Biotechnology 38, 1037–1043(2020)]. It would be informative to discuss these differences.

Response to reviewers' comments (Comments in black):

REVIEWERS' COMMENTS

Reviewer #1 (Remarks to the Author):

The authors have made an exceptional effort to include new data in the manuscript, which I believe provides greater context for the efficacy of their new editors in diverse genomic contexts.

My only remaining (minor) comments involve the interpretability of the color bars in the heatmaps across figures 1,2, and 4 (which appear to still be the relative 'activity' measure) compared to Figure 6, which shows the allele frequency.

Minimally, the authors should provide labels in the figures and potentially the legends to describe these values. That said, providing all data in units of the observed allele frequencies seems the most intellectually honest for all panels and would be more in line with prior work in this space. I'd recommend the authors update these unless they have a good reason to continue utilizing the current presentation.

Responses: Thanks a lot for the reviewer's suggestions, and we have provided the observed allele frequencies in the revised manuscript for Fig. 1c, Fig. 2. In Fig. 4a, the heat map represents the relative activity at different adenine positions for selected ABE-nSpCas9-DS variants, and we hope this heat map may provide guidance for appropriate choice of ABE variants. Based on the above considerations, we did not modify the current presentation, and which is also in consistent with previous studies (Fig. 1d, h and Fig. 2e,f; Huang TP, *et al.* Circularly permuted and PAM-modified Cas9 variants broaden the targeting scope of base editors. *Nat Biotechnol* **37**, 626-631 (2019)).

Reviewer #2 (Remarks to the Author):

The authors have mostly answered the issues I raised in the earlier review. At this time, I strongly feel that the strategy of using SpCas9-DS variants largely affect both Cas9 nuclease and adenosine deaminase. For example, the SpCas9-mediated DNA cleavage activities (Supplementary Figure 2) are not correlated (rather, negatively correlated) with the ABE-mediated base editing activities. i.e. ABEs from 770-TadA*-TadA* to 919-TadA*-TadA* did not generate indel mutations but were very effective for base editing. The canonical SpCas9-mediated DNA cleavage activities are positively correlated with the ABE/CBE-mediated base editing activities [Figure 1d at Nature Biotechnology 38, 1037 - 1043(2020)]. It would be informative to discuss these differences.

Responses: Thanks a lot for the reviewer's suggestions, and we have discussed the differences in the revised manuscript as follows: “Additionally, modest positive correlations have been observed between N-terminal-based conventional ABE/CBE activities and DNA cleavage activity of Cas9²⁴ while ABE-nSpCas9-DS770 to ABE-nSpCas9-DS919 described here maintained high editing

activities without any DNA cleavage activity. As regions 770-919 mainly cover the HNH domain in Cas9, it is possible that deaminase inside here might induce specific conformation changes to promote base editing process.”